# Patient and Clinical Factors at Admission Affect the Levels of Neutralizing Antibodies Six Months after Recovering from COVID-19

**DOI:** 10.3390/v14010080

**Published:** 2022-01-02

**Authors:** Xinjie Li, Ling Pang, Yue Yin, Yuqi Zhang, Shuyun Xu, Dong Xu, Tao Shen

**Affiliations:** 1Department of Microbiology and Infectious Disease Center, School of Basic Medical Sciences, Peking University, Beijing 100191, China; xinjieli@hsc.pku.edu.cn (X.L.); yuey@bjmu.edu.cn (Y.Y.); zhangyuqi_@bjmu.edu.cn (Y.Z.); 2Department and Institute of Infectious Disease, Tongji Hospital, Tongji Medical College, Huazhong University of Science and Technology, Wuhan 430030, China; pangling000@163.com; 3Department of Respiratory and Critical Care Medicine, Key Laboratory of Pulmonary Diseases of Health Ministry, Tongji Hospital, Tongji Medical College, Huazhong University of Science and Technology, Wuhan 430030, China; sxu@hust.edu.cn

**Keywords:** coronavirus disease 2019 (COVID-19), severe acute respiratory syndrome coronavirus (SARS-CoV-2), neutralizing antibody (NAb), diabetes, corticosteroids

## Abstract

The rate of decline in the levels of neutralizing antibodies (NAbs) greatly varies among patients who recover from Coronavirus disease 2019 (COVID-19). However, little is known about factors associated with this phenomenon. The objective of this study is to investigate early factors at admission that can influence long-term NAb levels in patients who recovered from COVID-19. A total of 306 individuals who recovered from COVID-19 at the Tongji Hospital, Wuhan, China, were included in this study. The patients were classified into two groups with high (NAb^high^, *n* = 153) and low (NAb^low^, *n* = 153) levels of NAb, respectively based on the median NAb levels six months after discharge. The majority (300/306, 98.0%) of the COVID-19 convalescents had detected NAbs. The median NAb concentration was 63.1 (34.7, 108.9) AU/mL. Compared with the NAb^low^ group, a larger proportion of the NAb^high^ group received corticosteroids (38.8% vs. 22.4%, *p* = 0.002) and IVIG therapy (26.5% vs. 16.3%, *p* = 0.033), and presented with diabetes comorbidity (25.2% vs. 12.2%, *p* = 0.004); high blood urea (median (IQR): 4.8 (3.7, 6.1) vs. 3.9 (3.5, 5.4) mmol/L; *p* = 0.017); CRP (31.6 (4.0, 93.7) vs. 16.3 (2.7, 51.4) mg/L; *p* = 0.027); PCT (0.08 (0.05, 0.17) vs. 0.05 (0.03, 0.09) ng/mL; *p* = 0.001); SF (838.5 (378.2, 1533.4) vs. 478.5 (222.0, 1133.4) μg/L; *p* = 0.035); and fibrinogen (5.1 (3.8, 6.4) vs. 4.5 (3.5, 5.7) g/L; *p* = 0.014) levels, but low SpO_2_ levels (96.0 (92.0, 98.0) vs. 97.0 (94.0, 98.0)%; *p* = 0.009). The predictive model based on Gaussian mixture models, displayed an average accuracy of 0.7117 in one of the 8191 formulas, and ROC analysis showed an AUC value of 0.715 (0.657–0.772), and specificity and sensitivity were 72.5% and 67.3%, respectively. In conclusion, we found that several factors at admission can contribute to the high level of NAbs in patients after discharge, and constructed a predictive model for long-term NAb levels, which can provide guidance for clinical treatment and monitoring.

## 1. Introduction

The severe acute respiratory syndrome coronavirus type 2 (SARS-CoV-2) pandemic has caused havoc around the world. Immunity after recovery from Coronavirus disease 2019 (COVID-19), is currently a subject of discussion in efforts to combat the pandemic. Persistent high levels of protective antibodies in individuals who recover from COVID-19 are thought to guard against reinfection from the SARS-CoV-2 [1,2]. However, due to differences in infection conditions, treatment regimens, and individual immune status, the production and duration of protective antibodies often tend to be poles apart among recovered patients.

Several studies have confirmed that the protective antibodies, especially neutralizing antibodies (NAbs) against SARS-CoV-2, rapidly decline within a few months after recovery from the disease, risking some patients at the edge of reinfection [3,4,5,6]. Interestingly, the rate of decay and decline in protective antibodies is highly heterogeneous across individuals [7,8]. The levels of neutralizing antibodies six months, or even more, after recovery from COVID-19, remain high in a few individuals, which can help them respond rapidly to prevent reinfection. A study that investigated recovered patients who were infected with SARS-CoV-2 in the early stages, have reported that at least 90% of convalescents retained positive NAbs and SARS-CoV-2-specific T-cell responses, 6 and 12 months after the disease onset, although varying in degree [9]. In addition to the immune memory characteristics of survivors, factors during early hospitalization associated with the persistence of high levels of NAb in patients six months or longer after recovery, are still infancy and, thus, worth exploring. Given the social and economic implications of the pandemic, estimating the efficacy and duration of long-term protective antibodies after discharge from hospitals, based on early indicators, is attractive but challenging. It can also facilitate appropriate medical treatment and care in the future.

In this study, we investigate the demographic and clinical factors at admission, or the early stage of patients’ hospitalization, associated with prolonged high levels of NAb against SARS-CoV-2. Data for patients that recovered from COVID-19 for at least six months were analyzed. Additionally, a model for predicting the long-term levels of NAb against COVID-19 after recovery was also constructed using the Gaussian mixture model. This model helps to guide treatment strategies and monitor responses to COVID-19 therapy, and infer long-term antibody protective efficacy from the earliest indications of hospitalization.

## 2. Materials and Methods

### 2.1. Study Population

A total of 306 individuals who recovered from COVID-19 were enrolled in this study. As described in the previous study [10], these patients were hospitalized and discharged from Tongji Hospital of the Huazhong University of Science and Technology, Wuhan, China, during 2020, as a result of laboratory-confirmed COVID-19. Given that the hospital is a local designated hospital for severe and critical illnesses, the patients had suffered moderate, severe, to critical COVID-19 infection. The disease severity was assessed at admission according to the “Chinese management guideline for COVID-19 (version 7.0)” [11]. All the convalescents included in this study were discharged from the hospital for more than six months, and none were exposed to the SARS-CoV-2 virus or suffered reinfection during the follow-up period. Simultaneously, cases included also met (1) age ≥ 18 years, (2) non-history of major medical or surgical conditions, such as malignant carcinoma (liver cancer, lung cancer, and so on), or organic transplantation and (3) non-psychiatric conditions, and were available for follow-up and evaluation.

Patients’ categories of severity were defined as follows: moderate: patients diagnosed with COVID-19 present with fever and respiratory symptoms, and pneumonia manifestations visible via imaging. Severe: patients diagnosed with COVID-19 met any of the following criteria: (1) respiratory distress with RR ≥ 30 times/min; (2) peripheral oxygen saturation (SpO_2_) ≤ 93% at rest; and (3) arterial partial pressure of oxygen (PaO_2_)/fraction of inspired oxygen (FiO_2_) ≤ 300 mmHg (1 mmHg = 0.133 kPa). Critical: meeting any of the following: (1) respiratory failure, requiring mechanical ventilation; (2) shock; (3) other organ failures, requiring intensive care unit (ICU) monitoring; or (4) death.

### 2.2. Data Collection

Patient data collected at baseline included: (1) demographic characteristics (age, gender, and so forth); (2) time from onset of illness to hospital admission, length of hospital stay, and disease severity; (3) clinical signs and symptoms at admission, underlying comorbidities, and treatments regimen; and (4) findings for clinical and biochemical tests at the early stage of hospitalization (initial systematic examination and comprehensive assessment of the COVID-19 patient at admission, usually within three days of hospitalization), as well as the physiological status of the patients, which were extracted from the patient’s electronic medical records. Furthermore, the patient-related indicators in (1), (2), and (3) were generalized as “patient factors”.

### 2.3. Classification of Patients

Six months after patients were discharged from the hospital, blood samples were collected from the recovered patients and assayed for NAb levels. The patients were classified into four groups based on the level of NAb against COVID-19. First, patients were classified into the NAb^high^ (high levels of NAb, *n* = 153) and NAb^low^ group (low levels of NAb, *n* = 153), based on the median level of NAb against COVID-19. Furthermore, patients in the fourth quartile (top 25% of the NAb levels) and the first quartile (the bottom 25% of the NAb levels) were further classified into the NAb^higher^ (*n* = 76) and NAb^lower^ group (*n* = 76), respectively (Figure 1).

### 2.4. Neutralizing Antibody Assay

To evaluate the level of NAb against COVID-19, the blood samples of COVID-19 convalescents were collected and centrifuged with the assistance of a medical professional. The extracted plasma was stored at 4 °C and analyzed within 24 h. Samples that could not be analyzed within this period were stored at −80 °C and assayed within one week. The iFlash-2019-nCoV NAb kit (YHLO, Shenzhen, China, Cat: C86109) and the full-automatic chemiluminescent analyzer (iFlash 3000) were applied to assess the level of SARS-CoV-2 NAbs in plasma samples. This approach was a one-step competitive strategy chemiluminescent immunoassay (CLIA) for the quantitative detection of NAb that blocks the binding between the receptor-binding domain (RBD) and angiotensin-converting enzyme 2 (ACE2). According to the manufacturer’s instructions, briefly, the plasma of samples was firstly incubated with the SARS-CoV-2 RBD antigen-coated paramagnetic microparticles. If the plasma sample contained NAb against the antigens, an antigen–antibody complex forms. The ACE2 protein acridine ester marker was then added to competitively bind the remaining RBD antigens, forming a bead-coated reaction complex. Upon introducing a magnetic field, the micro-magnetic particles were adsorbed to the reaction tube wall, but the unbound materials were washed away by the detergent. A chemiluminescent substrate was added to the immunoreactive complex, and the relative luminescence intensity (RLU) detected was inversely proportional to the number of NAbs in the plasma, which was automatically calculated and determined using the calibration curve. In particular, ≥10 AU/mL indicated a positive result of NAb. The superior sensitivity and specificity of this method have been validated in several studies [12,13,14].

### 2.5. Model for Predicting Levels of COVID-19 NAb

The model for predicting long-term levels of COVID-19 NAb was developed using the machine learning method of the Gaussian mixture model. After comparing the differences between the NAb^high^ group and the NAb^low^ group, factors with relative significant distinction (*p* < 0.1), namely the type of therapy received (corticosteroids therapy and intravenous immunoglobulin (IVIG) therapy); diabetes comorbidity as well as pulse oxygen saturation (SpO_2_); lactate dehydrogenase (LDH) level; urea; C-reactive protein (CRP); procalcitonin (PCT); fibrinogen; and serum ferritin (SF) levels, were incorporated in the model. For the accuracy of the model, SF was not included in the model because data for many patients were missing. In addition, gender, age, and disease severity (classified as severe or above and non-severe) were also included to calibrate the model. A total of twelve candidate variables were incorporated into the model. Before modeling, continuous clinical variables were dichotomized according to the optimum cutoff value, by using the receiver operating characteristic (ROC) analysis (Appendix A). The Gaussian mixture model (GMM) categorizes variables based on the hierarchical clustering of models, which features sound clustering performance and is a feasible screening method. As an unsupervised clustering, the Gaussian mixture model allows an intuitive observation of the distribution model under different combinations. Briefly, it was assumed that Gaussian distributions existed in the collected data and each distribution represented a cluster. Data points of the same distribution were first grouped together. The new probability for each data point was then assessed, followed by iterative re-classification. The highest rank in the optimal clustering would be selected after repeated training. The relationship between the various factors and levels of NAbs was assessed using univariate and multivariate regression analyses. ROC curves with AUC were constructed to assess the predictive validity of the model. Data were analyzed using the mclust package of R software (version 4.0.1).

### 2.6. Statistical Method

Differences between groups for categorical variables expressed as counts and percentages were analyzed using the χ2 test or Fisher’s exact test, as appropriate. Continuous variables were expressed using medians and inter-quartile range (IQR), and were analyzed using the Mann–Whitney U test. Statistical significance was set at two-tailed *p* < 0.05. Data were analyzed using SPSS 26.0 (IBM Corp., Armonk, NY, USA) and R software.

## 3. Results

### 3.1. Factors at Admission Associated with NAb^high^ and NAb^low^

Among the 306 study participants who recovered from COVID-19, 138 cases were males (45.1%). The convalescents were predominantly middle-aged and elderly persons, with a median (IQR) age of 62 (53, 68) years. NAbs were detected in the majority of the individuals (300/306, 98.0%), 6 months after their discharge. The median concentration for the NAbs was 63.1 (34.7, 108.9) AU/mL. At admission, 56.6% of patients were moderately ill, 40.7% were severely ill, and only 2.7% were critically ill. The majority of patients presented with fever (84.1%) and cough (80.5%) on admission and were appropriately treated as needed (Appendix A).

Patients were divided into NAb^high^ and NAb^low^ groups, according to the median level of NAb (Figure 2e). To uncover the clinical indicators and factors associated with the persistence of high levels of NAb after hospital discharge, we compared those two groups of COVID-19 recovered patients across various parameters. Although there was a relatively high proportion of patients with severe and critical illness in the NAb^high^ group, no significant difference was found in the severity between the two groups (*p* = 0.06) (Figure 2a). It was observed that the patients in the NAb^high^ group were more likely to receive corticosteroids (38.8% vs. 22.4%, *p* = 0.002) and IVIG therapy (26.5% vs. 16.3%, *p* = 0.033) than the NAb^low^ group (Figure 2b,d). Moreover, compared to the NAb^low^ group, a substantially higher proportion of patients in the NAb^high^ group presented with underlying diabetes (25.2% vs. 12.2%, *p* = 0.004) (Figure 2c). Analysis of the physiological and biochemical test results revealed that the serum SpO_2_ levels (median (IQR): 96.0 (92.0, 98.0) vs. 97.0 (94.0, 98.0)%; *p* = 0.009) at admission were relatively low in the NAb^high^ group individuals, in contrast with urea (4.8 (3.7, 6.1) vs. 3.9 (3.5, 5.4) mmol/L; *p* = 0.017); CRP (31.6 (4.0, 93.7) vs. 16.3 (2.7, 51.4) mg/L; *p* = 0.027); PCT (0.08 (0.05, 0.17) vs. 0.05 (0.03, 0.09) ng/mL; *p* = 0.001); SF (838.5 (378.2, 1533.4) vs. 478.5 (222.0, 1133.4) μg/L; *p* = 0.035); and fibrinogen (5.1 (3.8, 6.4) vs. 4.5 (3.5, 5.7) g/L; *p* = 0.014) levels, which were significantly high (Figure 2f–k). The comparison between the groups regarding other parameters is shown in Appendix A.

### 3.2. The Relationship between Long-Term Serum NAb Levels and Clinical Indicators

To further investigate the correlation between long-term NAb levels against COVID-19 after recovery and the clinical indicators at admission, we compared the factors between individuals in the top 25% and the bottom 25% (NAb^higher^ and NAb^lower^ group) (Figure 3d). There was no significant difference in disease severity between the NAb^higher^ and NAb^lower^ groups (Figure 3a). Interestingly, we found that some clinical indicators still differed between these two groups. It was observed that compared with NAb^lower^ individuals, a higher proportion of patients in the NAb^higher^ group received corticosteroids therapy during hospitalization (45.8% vs. 21.6%, *p* = 0.002) (Figure 3b). As for comorbidity, a larger proportion of patients in the NAb^higher^ group experienced a history of diabetes at the time of admission than in the NAb^lower^ group (31.9% vs. 8.1%, *p* < 0.0001) (Figure 3c). Moreover, patients in the NAb^higher^ group displayed significantly higher levels of serum CRP (median (IQR): 24.4 (4.9, 90.3) vs. 8.5 (1.9, 32.8) mg/L; *p* = 0.003); PCT (0.07 (0.05, 0.12) vs. 0.05 (0.03, 0.07) ng/mL; *p* = 0.009); and fibrinogen (5.2 (3.8, 6.5) vs. 3.8 (3.3, 4.8) g/L; *p* < 0.0001), but lower SpO_2_ (96.0 (92.0, 98.0) vs. 97.0 (95.0, 98.0)%; *p* = 0.049) levels, relative to the NAb^lower^ counterparts (Figure 3e–h).

These findings suggest that corticosteroids therapy and diabetes comorbidity can promote the sustained production of NAb in patients who recover from COVID-19. Additionally, the acute inflammation-related factors, such as CRP, PCT, as well as fibrinogen, and the SpO_2_ levels in the initial stage of COVID-19 infection, influence the long-term production of NAb against the virus.

### 3.3. Model for Predicting Long-Term NAb Levels

For the establishment of the clinical predictive model of long-term NAb after recovery, logistic regression analyses were performed to assess the screened 12 candidate indicators. The logistic regression models for the 12 factors associated with consistently high levels of NAb had a total of 8191 formulas. Based on GMM, the 12 factors were divided into 7 clusters. After repeated training, the cluster with the highest AUC was selected for the prediction of the NAb levels of patients six months after discharge (Figure 4).

Overall, the developed predictive model consisted of 9 clinical indicators, including age; gender; disease severity; corticosteroids therapy; IVIG; and diabetes comorbidity, as well as the SpO_2_, urea, and CRP level, which were found to influence the NAb levels six months after recovery from COVID-19. The model displayed an average accuracy of 0.7117 for the GMM classifier (Figure 4). To validate the predictive effect of the combination indicators, the ROC analysis was conducted. The predictive model had an AUC value of 0.715 (0.657–0.772), whereas its specificity and sensitivity were 72.5% and 67.3%, respectively (Figure 5).

## 4. Discussion

The presence and level of SARS-CoV-2-induced neutralizing antibodies varied widely among recovered patients based on patient and treatment factors. It has been reported that serum NAb peaks within 3–5 weeks after SARS-CoV-2 infection. However, the titers and neutralizing activity decline rapidly within 1–6 months and, in some patients, the NAbs are completely undetectable several months after infection [15]. Surprisingly, in some convalescents, the titer and activity of the NAbs remain high and stable, respectively extensively beyond the follow-up period after recovery [16]. In this study, NAbs against SARS-CoV-2 were detected in 98.0% of study participants, 6 months after discharge from the hospital, despite individual differences in the levels. Earlier research also indicated consistent neutralizing activity in most subjects for as long as 6 months [17]. A study from Wuhan, China indicated that NAb concentrations in recovered patients were relatively stable for at least 9 months, regardless of whether they were symptomatic or not [8].

Existing studies suggest that NAb titers in COVID-19 survivors generally positively correlate with disease severity [18,19]. Nevertheless, in our cohort, NAb levels did not exhibit significant differences overall in patients of different disease severities (Appendix A). This discrepancy can be because all of the participants included in this study were all inpatients, who suffered from a moderate-to-critical illness. Lacking mild and asymptomatic patients for comparison, the difference in terms of disease severity was relatively small. Overall, numerous factors influence disease severity, and this complex phenomenon needs further exploration. In particular, when disease severity is similar in the infected population, the impact of clinical therapy, patient immune response, and other comorbidities during hospitalization are of concern. Diabetes comorbidity and corticosteroids therapy can contribute to high levels of NAb, six months after recovery from COVID-19. For diabetes patients, several speculations can help to explain this: (1) the disease is highly prevalent in the elderly with a natural susceptibility to COVID-19 and are likely to experience severe illness, resulting in a poor prognosis; (2) the persistent chronic inflammatory response induced by diabetes can amplify the immune response against SARS-CoV-2, resulting in more intensive and prolonged inflammatory responses that enhances the generation of memory T and B cells; and (3) the imbalance of coagulation and the fibrinolytic system in diabetic patients impairs the vascular endothelial function, which further affects the immune function and secretion of related factors. As no significant difference in disease severity was found between patients with diabetes and non-diabetes (Appendix A), the distinction of NAb levels could be due to diabetes itself.

In one study in Mexico, among 32,583 patients, diabetes was found to increase the risk of SARS-CoV-2 infection and the subsequent development of serious illness, and was related to inflammation and high mortality [20]. As a chronic inflammatory disease characterized by multiple metabolic and vascular abnormalities, diabetes promotes the production of tissue inflammation-mediated adhesion molecules and is linked with the acceleration and worsening of atherothrombosis. This increases advanced glycation end products (AGEs) and pro-inflammatory cytokines, further influencing immune responses to viral infections [21]. Pro-inflammatory cytokines, such as interleukin-6 (IL-6) and tumor necrosis factor-alpha (TNF-α) produced under viral infections, can exacerbate the severity of illness, resulting in poor prognosis, including cytokine storms.

In this study, we found that corticosteroid therapy affects the long-term production of neutralizing antibodies in different patients, but this phenomenon has not aroused concern. Corticosteroids are often widely used in the treatment of COVID-19 patients in ICU [22]. It inhibits the transcription and action of several cytokines, and modulates the proliferation, activation, differentiation, and survival of T cells and macrophages. Corticosteroids restrain the secretion of several pro-inflammatory cytokines produced by Th1 and macrophages, including IL-1β, IL-2, IL-6, and TNF-α. Patients subjected to corticosteroid therapy during hospitalization are usually in pressing need of improving clinical symptoms and oxygenation, and when the virus invades the lung epithelium, the organism is stimulated to activate specific immune cells, macrophages, and natural killer cells to produce abundant cytokines and chemokines [23]. The application of corticosteroids can alleviate chronic obstructive pulmonary disease (COPD) by modulating inflammation in the lungs [24]. Although corticosteroids can reduce the need for mechanical ventilation, they do not fully improve oxygenation in the body. Corticosteroid therapy can excessively suppress the functioning of the immune cell, thus delaying viral clearance and inducing the slow but continuous stimulation of T and B lymphocytes. Combined, these events sustain the production of numerous NAbs.

Many factors, including SpO_2_, urea, CRP, PCT, SF, and fibrinogen levels, varied substantially between patients with long-term high and low levels of NAbs, and this was closely related to hypoxia, inflammation, and coagulation disorder. These indicators have been linked with the severe disease of COVID-19, in previous studies [25,26]. Low SpO_2_ exacerbates pneumonic injury and the resultant hypoxemia is associated with poor clinical prognosis [27]. An immune response to COVID-19 increases the CRP, PCT, and SF, among other indicators, especially in critically ill individuals [25]. Elevated fibrinogen indicates the risk of systemic hypercoagulability and thrombotic microangiopathy in COVID-19 patients, which is inseparable from the thromboinflammation or immunothrombosis caused by COVID-19-related inflammation [28]. Blood urea levels reflect the renal function, and underlying systemic vascular and inflammatory complications. As such, it is a biomarker for inflammation-related complications [29]. Established studies also suggest that titers for NAb against SARS-CoV-2 are linked with CRP, implying that high levels of NAb can be associated with a strong inflammatory response [30,31]. In contrast, there are no significant differences in the levels of serum cytokines and chemokines between healthy individuals and asymptomatic patients. This can be because the low inflammatory responses in these individuals are insufficient to induce persistent immune responses, capable of maintaining prolonged high titers of NAb [32].

Currently, the SARS-CoV-2 is wreaking havoc across the world. However, the constant mutation of the virus strains and the rapid decay of antibodies after recovery, expose COVID-19 survivors to a plight of reinfection. An early estimate of a patient’s long-term antibody levels, based on indicators at admission, can help clinicians to promptly adjust strategies during treatment and after discharge, maximizing the maintenance of high levels of long-term protective antibodies and reducing the risk of reinfection. Applying this predictive model at an early assessment, provides a clinical reference for treatment decisions, inpatient care, and individualized programs for COVID-19 patients, effectively enabling target measures to enhance post-discharge antiviral resistance and immune protection.

Regarding the limitations, firstly, we were not able to monitor and compare ongoing changes in antibody levels in convalescents over multiple periods. Secondly, given the retrospective nature of the research, the test results of clinical indicators for some patients were incomplete when information was extracted. Lastly, the established prediction model requires further validation in a subsequently larger clinical population.

## 5. Conclusions

In conclusion, several factors at admission can contribute to a high level of NAbs in patients, six months after discharge. We found that diabetes comorbidity and corticosteroids therapy, as well as the SpO_2_, CRP, PCT, and fibrinogen levels, affect the prolonged production of NAbs against SARS-CoV-2 in individuals who recover from COVID-19. In addition, we constructed a predictive model for sustained NAb levels in convalescents. The findings of this study can provide some valuable guidance for the treatment and monitoring of patients in clinical recovery.

## Figures and Tables

**Figure 1 viruses-14-00080-f001:**
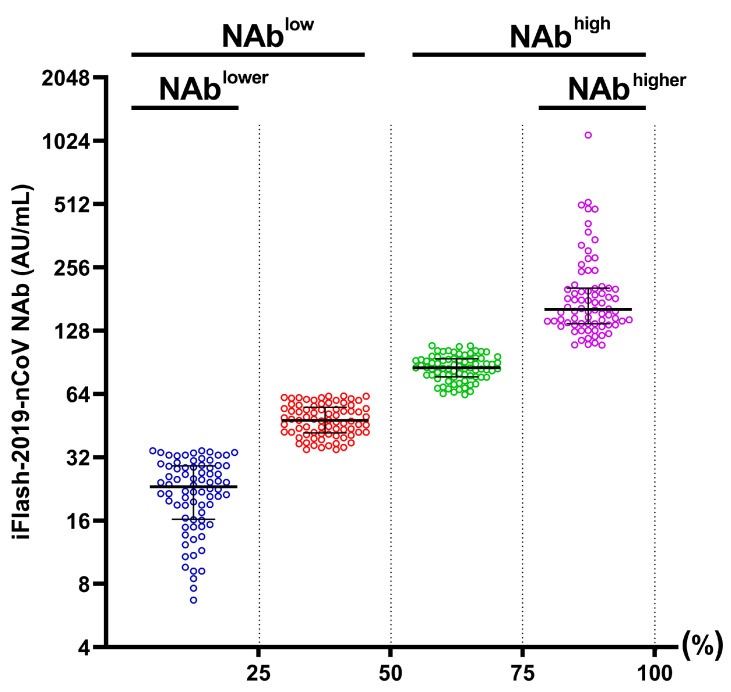
The level of SARS-CoV-2 NAb in 306 individuals 6 months after recovering from COVID-19. Patients were divided into NAb^high^ (median (IQR): 108.8 (85.0, 161.8) AU/mL) and NAb^low^ groups (34.9 (23.1, 48.1) AU/mL), based on the median NAb levels. In addition, 50% of individuals in the upper NAb^high^ and lower NAb^low^, were further classified into NAb^higher^ (155.3 (116.6, 200.7) AU/mL) and NAb^lower^ groups (23.2 (16.2, 30.1) AU/mL). NAb, neutralizing antibody.

**Figure 2 viruses-14-00080-f002:**
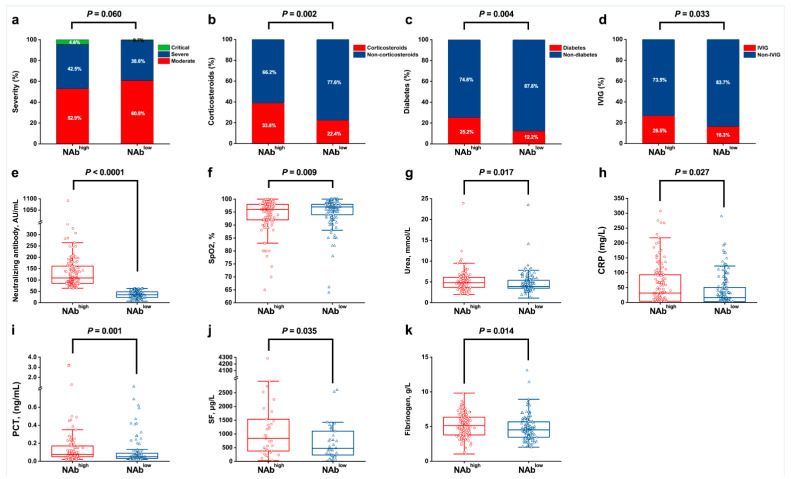
Various patient factors and clinical indicators between the NAb^high^ group and the NAb^low^ group at admission. (**a**), severity; (**b**), corticosteroids; (**c**), diabetes; (**d**), IVIG; (**e**), neutralizing antibody; (**f**), SpO_2_; (**g**), urea; (**h**), CRP; (**i**), PCT; (**j**), SF; (**k**), fibrinogen. NAb, neutralizing antibody; IVIG, intravenous immunoglobin; SpO_2_, pulse oxygen saturation; CRP, C-reactive protein; PCT, procalcitonin; and SF, serum ferritin.

**Figure 3 viruses-14-00080-f003:**
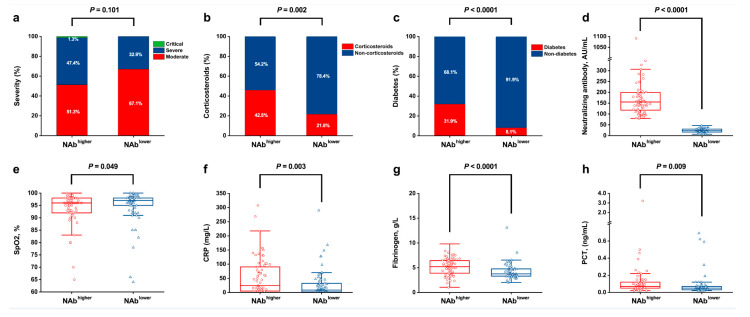
The difference in various indicators between the NAb^higher^ group and the NAb^lower^ group at admission. (**a**), severity; (**b**), corticosteroids; (**c**), diabetes; (**d**), neutralizing antibody; (**e**), SpO_2_; (**f**), CRP; (**g**), fibrinogen; (**h**), PCT. SpO_2_, pulse oxygen saturation; CRP, C-reactive protein; and PCT, procalcitonin.

**Figure 4 viruses-14-00080-f004:**
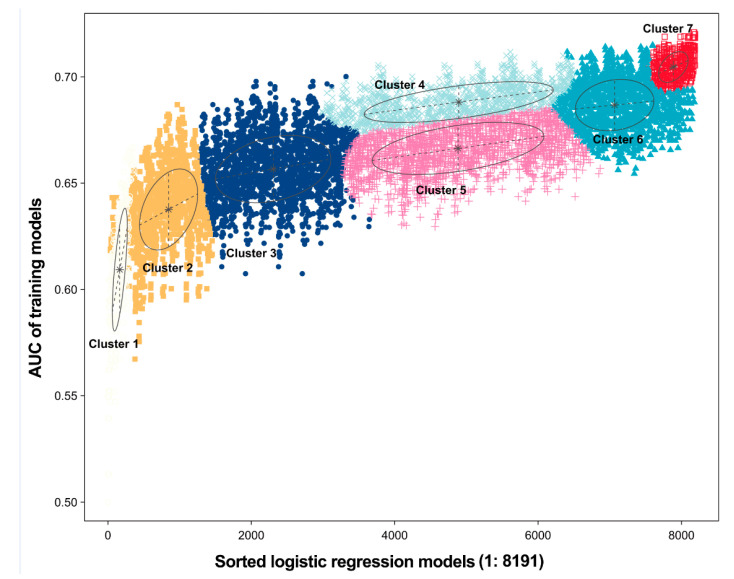
The logistic regression model correlated with the AUC scores, based on Gaussian mixture models for predicting the levels of NAb six months after recovery from COVID-19. There are 7 clusters of 8091 combinations; the optimal model has an average accuracy of 0.7117. Each color or shape represents a different cluster clustered by Gaussian clustering, and the horizontal coordinates represents the number of combinatorial models generated. The asterisks represent the core positions in the clusters, and the covariance matrices of the clusters bind together to form circular class clusters. The shape of the n-dimensional Gaussian distribution is determined by the covariance of each class cluster.

**Figure 5 viruses-14-00080-f005:**
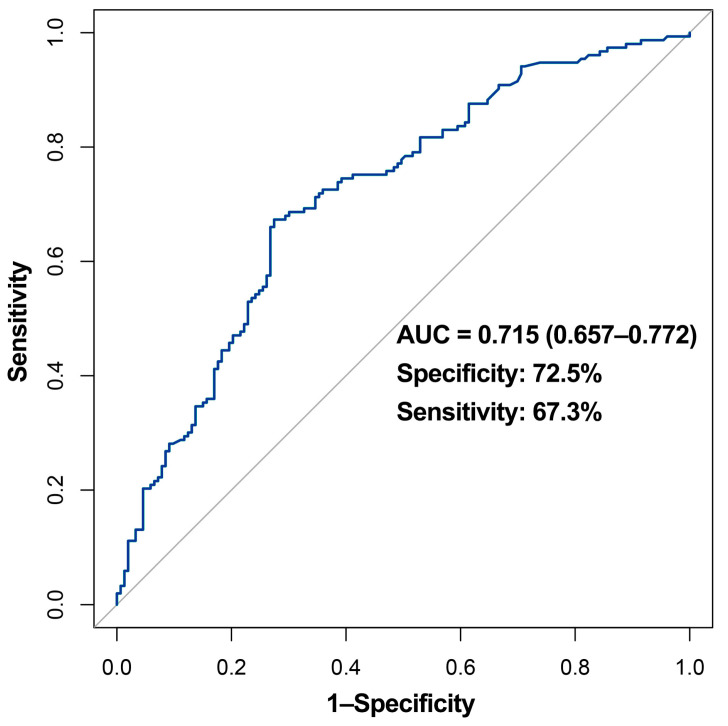
ROC curve for the model predicting the level of SARS-CoV-2 NAbs. The model incorporates the age and gender of the patients, the severity of the illness, therapy received (corticosteroids and IVIG), and diabetes comorbidity, as well as serum SpO_2_, urea, and CRP levels. The AUC for the predictive model is 0.715 (0.657–0.772), whereas its specificity and sensitivity are 72.5% and 67.3%, respectively. IVIG, intravenous immunoglobin; SpO_2_, pulse oxygen saturation; and CRP, C-reactive protein.

## Data Availability

The data used to support the findings in this study are available from the corresponding authors upon request.

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
