# Peer review of "Patient and Clinical Factors at Admission Affect the Levels of Neutralizing Antibodies Six Months after Recovering from COVID-19"

_viruses, 2022, doi:10.3390/v14010080_

Round 1

Reviewer 1 Report

Thank you for giving me an opportunity to review this paper. Authors investigated demographic and clinical factors at admission or at the early stage of patients’ hospitalization associated with prolonged high levels of NAb against SARS-CoV-2. The manuscript is well written. However, there are several points that need to be addressed.

  1. The title is not appropriate, why they have mentioned "several". it is not appropriate in the title.
  2. In the abstract, the study objective is missing.
  3. In the method parts, authors should explain the Gaussian mixture model. Why they have used this model? what is the advantages of using this model? Why they have used it instead of other models?
  4. In the discussion part, please provide more information about clinical implications.

Author Response

Q1: The title is not appropriate, why they have mentioned "several". it is not appropriate in the title.

A: Thank you very much for your suggestion. We have revised the title by removing "several" to promote a more accurate expression. Please see the new title in the revised manuscript.

Q2: In the abstract, the study objective is missing.

A: Thank you for your helpful suggestions. The objective of this study was to explore factors at admission that influence the long-term neutralizing antibody levels in patients recovered from COVID-19. Based on your kind reminder, we have complemented the abstract, please see the line21-line22 in the revised manuscript for more details.

Q3: In the method parts, authors should explain the Gaussian mixture model. Why they have used this model? what is the advantages of using this model? Why they have used it instead of other models?

A: Thank you very much for your instructive advice. Generically speaking, the Gaussian mixture model completes the clustering by selecting components to maximize the posterior probability. The method is calculated using an iterative algorithm that eventually converges to a local optimum, improving the stability and accuracy of the prediction model. In this study, we would like to explore whether the selected promising indicators have a certain impact or how they can be optimally combined under different combinations. The Gaussian mixture model, as an unsupervised clustering, allows an intuitive observation of the distribution model under different combinations, and in this respect, the model shows irreplaceable merits, which is why we have chosen this method. We have supplemented the methods with more information to enable a clearer understanding for readers. Please see the line154-line158 for detailed information in the revised manuscript.

Q4: In the discussion part, please provide more information about clinical implications.

A: Thank you very much for pointing this out. Based on your suggestion, we have added more information about clinical implications and made specific elaborations in the section of the discussion. Please see the revised manuscript of line348-line356 for changes.

Reviewer 2 Report

File attached

Author Response

We are grateful for the constructive comments from your letter, which have helped us tremendously to improve the quality and clarity of the paper. All the authors have seriously discussed all these comments and have tried our best to modify our manuscript to meet the requirements of your journal. We believe the points emphasized by the reviewers are addressed in this revised manuscript. All the revisions are highlighted in red in the revised manuscript and supplementary materials. Point-by-point responses are attached below.

Thank you very much for your consideration. We appreciate your help and patience. We hope that the revisions are satisfactory but should there be any other changes that are necessary to improve the paper, please feel free to contact us.

General Comments

Q1: The manuscript is poorly written with many spelling mistakes and grammatical errors. The main premise of the article is identification of markers at the time of admission that affect neutralizing antibody (NAb) titres through predictive modeling. The concept is good, but the authors only consider every patient factor as a direct consequence of neutralizing antibody titres, which is not correct. Eg. The authors correlate high NAb titre with severe disease, use of corticosteroids and IVIG therapy. But the authors should consider that severe disease is the primary cause of high NAb titres and requires the use of more potent medication and drop in body vitals. The discussion part of this article focuses too much of the linking of disease severity to other parameters inconsideration, which is not done during the analyses of study finding. The discussion part needs to focus more on the central idea of the article which is, reason why high/ low NAb titers will cause the other parameters to increase or decrease.The authors have done a lot of analyses, but they have to explain them in more depth. A major drawback of the study is the collection of samples at 6 months post infection for determination of NAb titres. Acute phase samples for the same patients would have further validated the findings.

A: We sincerely appreciate your thorough review of our manuscript and detailed and insightful comments. We are also encouraged by your constructive comments on the usefulness and novelty of our research. These valuable suggestions were carefully considered by the whole working group and, after active discussion, we made every effort to address them, as shown below. Following the instructive suggestions, we conducted extensive revision work related to the inadequacies of the manuscript, including refining the methods and results, reorganizing the structure of the article, adding and citing literature as appropriate, as well as touching up the language and rechecking the grammar.

Abstract

Q2: Line 23: Explain the abbreviation before use and also use consistent and correct abbreviations.

A: Thank you so much for your careful checks. It was a minor formatting error in the typo, and we have corrected the abbreviation promptly and added an explanation after the initial abbreviation. We have also double-checked the subsequent formatting to avoid unnecessary missteps. Please see the line24-line25 of the new manuscript for specific corrections.

Q3: Line 28-29: Percent values between High NAb vs Low NAb groups will be helpful.

A: Thank you for your valuable suggestion. We have added a comparison of the various indicators between the NAbhigh and NAblow groups in this section of the abstract to help readers understand the results clearer. Please see the line30-line34 in the revised manuscript.

Introduction

Q4: Line 40: Please check the entire article for spelling mistakes and grammatical errors.

A: Thank you very much for your careful examination and friendly reminder. The manuscript has been polished by an English language editing company to improve readability. We have also carefully checked the entire manuscript to correct grammar/typo errors. Please see the markings and corrections in the revised manuscript.

Q5: Line 47: The authors refer studies where there is rapid decline in antibodies against SARS-C-V-2, but there are several studies that report long term antibody responses.

A: Thank you for your suggestion. In the introduction, in addition to reports of rapid declines in SARS-CoV-2 antibodies, there are indeed a handful of studies reporting long-term antibody responses, and we have described relevant researches in the subsequent of the article (please see the line55-line62 in the revised manuscript for details). It was the heterogeneity demonstrated by different patients in the long-term antibody response that motivated us to explore the admission factors that influence long-term neutralizing antibody levels.

Q6: Line 59: The authors should consider using the word pandemic instead of epidemic.

A: Thank you for your thoughtful examination. We have replaced "epidemic" with "pandemic" in the sentence to make it more accurate. Please see the line64 in the revised manuscript.

Materials and Methods

Q7: Line 76-78: If the hospital is a designated hospital for severe and critical illnesses, how were moderately symptomatic patients admitted.

A: Thank you for raising this concern. We are delighted to explain that this hospital was designated as the local hospital for the treatment of severe and critically ill patients when the pandemic outbreak in December 2019. This means that serious patients were prioritized for treatment here, but it does not mean that the hospital didn’t admit patients with moderate symptoms. During that time, a large number of COVID-19 patients with moderate illness were admitted to the hospital.

Q8: Line 79: Briefly describe the categorization of moderate, severe and critical symptomatic patients.

A: Thank you for your helpful advice. The disease severity of the patients included in this study was critically assessed at admission according to the "Chinese management guideline for COVID-19 (version 7.0)". In the revised manuscript, we have supplemented standards for the classification of moderate, severe and critical patients to complete the methodology. Please see the line91-line98 in the revised manuscript for detailed changes.

Q9: Line 83: The authors mention that there was non-history of major medical or surgical conditions, such as carcinoma amongst the convalescents. But 2.7% of study population had carcinoma mentioned in supplementary table 2. Please explain.

A: Thank you for noting this and providing suggestions. To prevent the unnecessary effect of substantial medical histories on recovered patients, we have excluded cases with significant medical comorbidities at the early stage of case enrolment (malignant tumors such as liver cancer, lung cancer) or major surgeries (including organ transplantation). However, regarding some patients with benign tumors or cancers that had been eradicated before admission (such as thyroid cancer or early-stage prostate cancer), the disease histories of these cases would not have a significant impact on the study results and were therefore included in the study cohort. We are aware that the presentation of this part could be ambiguous, we have revised the description in the methods and in the supplementary materials to avoid confusion. Please see the line88-line89 in the revised manuscript and the line6-line14 in the revised supplementary materials.

Q10: Line 90-91: Define “Early stages of hospitalization”

A: Thank you for your instructive advice. The electronic record of the patient and the relevant information were extracted from the initial systematic examination and comprehensive assessment of the COVID-19 patient at admission, usually within three days of hospitalization. In the new version of the manuscript, we have refined the definition of the “early stages of hospitalization” to facilitate better understanding. For detailed changes, please refer to the line104-line105 in the revised manuscript.

Q11: Line 95-100: What is the quantitative threshold between high and low NAb titres and what criterion was used to define high and low titres. Also please explain what is the fouth upper and lower quartile.

A: Thank you very much for your careful review. In this study, quantification of NAb titers was performed by one-step competitive strategy chemiluminescent immunoassay (CLIA), and according to the manufacturer's instructions, ≥10 AU/mL indicated a positive result of NAb. The classification of the NAbhigh and NAblow groups was based on the median NAb levels six months after discharge for all recovered individuals included in this study. Patients with NAb levels above 50% were defined as the NAbhigh group and those below 50% were defined as the NAblow group, while patients in the top 25% of the NAb levels were defined as the NAbhigher group and those in the bottom 25% were defined as the NAblower group. Thus, the fourth quartile (patients in the upper quartile) means the highest 25% of numbers, and the first quartile (patients in the lower quartile) means the lowest 25% of numbers. To avoid misunderstandings, we reorganized the text of this sentence to clearly define our groups. For the revised definition, please see the line114-line117 in the revised manuscript.

Q12: Line 127: For significant distinction, the p value has to be below 0.05. Could the authors justify the use of p<0.1 here.

A: Thank you very much for posing this question. In the ordinary statistical tests, p<0.05 is indeed the sign of a significant difference, and this is valid for the majority of our presented results. However, in constructing clinical prediction models, including multivariate logistic regression, when only variables with p<0.05 included in the univariate analysis are included, certain important factors likely have no chance to enter the multivariate model (such as age, gender, and other indicators that do have an impact on clinical outcomes). Therefore, at the screening step of the promising variables, the test level is often appropriately loosened and adjusted, for example, to include indicators with p<0.1 or 0.2 and to incorporate clinically relevant indicators to prevent the omission of important factors, rather than just discarding statistically nonsignificant variables. This approach has also been frequently mentioned in previous studies. Please see doi: 10.21037/atm.2019.08.63; doi: 10.1136/heartjnl-2016-310210 for more information.

Q13: Line 134-136: Please provide the individual ROCs used to dichotomize the clinical variables.

A: Thank you very much for your helpful suggestions. In the supplementary materials, we have attached the cutoff values for the dichotomous categories in the ROC analysis for the continuous variables in the clinical variables, and this data can make it easier for readers to understand our results. Please see the new Supplementary Table 3 (line19-line21 in the revised supplementary materials) and the line153 in the revised manuscript.

Statistical method

Q14: Line 147: How was GMM and ROC calculated?

A: Thank you very much for your comments. In the methods section, we have mentioned that GMM and ROC are working with a package called mclust for R software. mclust is a contributed R package for model-based clustering, classification, and density estimation based on finite normal mixture modelling. It provides functions for parameter estimation via the EM algorithm for normal mixture models with a variety of covariance structures and functions for simulation from these models. Also included are functions that combine model-based hierarchical clustering, EM for mixture estimation and the Bayesian Information Criterion (BIC) in comprehensive strategies for clustering, density estimation and discriminant analysis. Additional functionalities are available for displaying and visualizing fitted models along with clustering, classification, and density estimation results. Please see the line163-line165 in the revised manuscript.

Results

Q15: Line 157: Do the numbers in the brackets represent IQR? If so please mention accordingly.

A: Thank you very much for your kind reminder. The numbers in the brackets do represent the IQR for age, about which we have added a note in the revised manuscript. Please see the line176 in the revised manuscript.

Q16: Line 164-167: What is the correlation (r) between high NAb titres and disease severity? As has been widely reported, NAb titres are known to be higher in individuals with severe disease. The author mentions that the patients in NAbhigh group was more likely to receive corticosteroids and IVIG therapy than the NAblow group. This line gives a false interpretation that all individuals with high NAb titres require more potent medication to recover. These individual who have high NAb are given corticosteroids and IVIG therapy because of the severity of their disease.

A: In our cohort, compared to the NAblow group, despite the relatively high proportion of patients with severe and critical illness in the NAbhigh group, there was indeed no significant difference in severity between the two groups (p = 0.06), as presented in the Figure 2a and Supplementary Table 1 in the supplementary materials. In addition, we have performed the correlation analysis between NAb titers and disease severity, but we didn’t find a significant correlation between them (Spearman’s correlation coefficient: r = 0.124; p = 0.030, data not shown). A point should be underlined that we did not deny the correlation between high NAb and disease severity, but possibly because the majority of patients had a severity above moderate, there was relatively little variation in disease severity between cases. Therefore, in terms of disease severity, patients in different NAb groups did not present significant differences. In addition, we need to clarify that not only severe patients were treated with corticosteroids and IVIG. The application of therapies such as high-flow nasal cannula oxygen therapy (HFNC), non-invasive mechanical ventilation (NIV) and invasive mechanical ventilation (IV), and continuous renal replacement therapy (CRRT) were more indicative of a severe condition of disease compared to the above two interventions, but these therapies did demonstrate a significant correlation with high levels of NAb, with no significant difference between the two groups (please see the Supplementary Table 1). This suggests, to some extent, that the maintenance of high levels of long-term NAb in the NAbhigh group may have been influenced by their previous treatment with corticosteroids or IVIG.

Q17: Line 170-174: Please add average/ GMT values along with p values for serum SpO2, urea, CRP, PCT, SF and fibrinogen levels.

A: Thank you for your instructive suggestions. We have added data and p values for serum SpO2, urea, CRP, PCT, SF and fibrinogen levels in the revised manuscript, and since the data do not conform to a normal distribution, the data are expressed as median (IQR). Please see the line193-line198 and the line212-line216 in the revised manuscript.

Q18: Line 187: Please provide p value for SpO2 levels.

A: Thank you for noting this point. We apologize for this small slip in the draft and the p values for SpO2 levels have been added in time in the revised manuscript, please see the line215-line216 in the revised manuscript.

Q19: Line 188-192: Again, the longevity of NAb may not be because of diabetes, but is most likely because severe disease produces higher titres of NAb.

A: Thank you very much for your friendly caution. To complement the proof of this, we have added a comparison of disease severity between diabetic and non-diabetic patients to the results (please see the Supplementary Figure 1 in the revised supplementary materials). In this study cohort, no significant difference in disease severity was observed between diabetic and non-diabetic patients, while long-term Nab levels showed different levels, which could explain, to some extent, that this difference in long-term NAb levels could be related to diabetes comorbidity rather than differences in disease severity in this cohort.

Q20: Line 212 Figure 1: Please provide median/mean/ GMT in AU/mL for each of the categories represented.

A: Thank you for your helpful suggestions. We have added the median and IQR for each category in the figure legend of Figure 1 to make the data clearer and more intuitive. Please see the line237-line241 in the revised manuscript for details.

Q21: Line 217 Figure 2: It will be more clear to the reader if each sub-image was labeled as a, b, c…. etc and reference for them were provided in the main text.

A: Thank you for your valuable suggestions. Following this suggestion, we have labeled each sub-image in Figure 2 and Figure 3 and supplemented the description in the results to provide a reference for better understanding. Please see the revised new Figure 2 and new Figure 3 and the line182-line216 in the revised manuscript.

Q22: Line 226 Figure 4: As this is an important image on the manuscript, please clearly define this image. What do the circle, dotted line and star represent? Define what each cluster means and how was it divided.

A: Thank you for raising this point. The selected variables were classified by model-based hierarchical clustering via Gaussian mixture model (GMM), which has good cluster performance. Each color or shape represented a different cluster clustered by Gaussian clustering, where the horizontal coordinates were from 1 to 8191 (representing the number of combinatorial models generated). To better distinguish the representatives of each model, we further divided the variable combination of data distribution, with the asterisks representing the core positions in the clusters and the covariance matrices of the clusters bound together to form circular class clusters. The shape of the n-dimensional Gaussian distribution was determined by the covariance of each class cluster. We have noted the specific meaning represented by the circles, dotted lines, and stars in the image in the legend to Figure 4, and have added the meaning of each cluster and the basis for the division to provide clarity to readers. Please see the line255-line260 in the revised manuscript for details.

Q23: Line 234 Figure 5: The ROC is very unclear. How do the authors represent all the factors (age, gender etc.) in one ROC? Also what parameter acts as gold standard to determine the sensitivity and specificity?

A: Thank you very much for your suggestion. In this study, for the selected indicators, to verify whether the combination of these indicators meets the gold standard in a statistical significance, we constructed a generalized linear model using the GLM function to obtain the final model and observed the accuracy and sensitivity of this model at high and low levels of antibody classification using ROC analysis. Thus, this ROC curve characterizes not a single metric, but validation and assessment of the effectiveness of the whole model. This similar ROC analysis curve is also demonstrated in the doi: 10.1136/heartjnl-2016-310210 article for reference.

Discussion

Q24: Line 237-238: Authors have not acknowledged the variation in NAb titres due to disease severity. Aslo what are patient factors?

A: Thank you very much for your advice. We have not denied that Nab titers vary by disease severity throughout, but since in this study we are focusing on relevant factors at admission that influences long-term NAb after discharge, the effect of a single indicator of disease severity could not be so obvious in this cohort. We believe it’s a comprehensive factor influenced by multiple contributors. In addition, "patient factors" refers to generalizations of patient-related personal indicators, such as demographic characteristics (age, gender, and so forth.); 2) time from onset of illness to hospital admission, length of hospital stay, and disease severity; 3) clinical signs and symptoms at admission, underlying comorbidities, etc. For clarity, we have added additional statements in the revised manuscript, please see line107-line108 in the revised manuscript.

Q25: Line 250-253: Authors have not shown comparison of NAb levels based on disease severity. So proof is needed to say that “NAb levels did not exhibit significant differences overall in patients of different disease severity”.

A: Thank you very much for your comments. To demonstrate that "NAb levels were not significantly different overall in patients with different disease severity", we added a comparison of NAb levels in patients with different disease severity (please see the Supplementary Figure 2 in the revised supplementary materials and the line286-line287 in the revised manuscript). This result indicated that no significant differences were found in neutralizing antibody levels among patients with different disease severity in this study. But in our opinion, disease severity itself could be associated with a variety of complex factors and other indicators may be a multifaceted and comprehensive reflection of disease severity. Therefore, we did not deny the possible influence of disease severity on NAb levels.

Q26: Line 278-294: The authors extensive talk about the use of corticosteroids to treat severe/critical patients and its effect on the immune system, but themselves have not focused on impact of disease severity on antibody production, need for higher potency medication and drop in body vitals. Also corticosteroids suppresses certain factors of the immune system, they however do not significantly affect antibody production (How corticosteroids work; H N Claman; PMID: 803519 DOI: 10.1016/0091-6749(75)90010-x). Even if use of corticosteroids delays viral clearance and thus provides slow but continuous stimulation of T and B lymphocytes, it does not give explanation of high NAb titres. It just gives proof for presence of NAb for a long time post infection.

A: Thank you very much for your detailed and insightful suggestions. We have explained the effect of disease severity on antibody production in the previous question, and although our results do not show a significant correlation between the single indicator of "severity" and long-term levels of NAb, we do not deny the effect of disease severity on antibody levels. We believe that many other relevant indicators could also be a combined presentation of disease severity, which in turn may have an impact on long-term antibody levels. Therefore, we have highlighted this point in the discussion. As for the correlation between high levels of NAb and corticosteroids, we are interested in this phenomenon. However, due to the scarcity of available studies, it is difficult to conclude the exact mechanism of the association between this therapy and high levels of long-term NAb. We hope that this interesting phenomenon would receive more attention and be discussed and validated in more depth in subsequent studies.

Q27: Line 295-306: The authors again talk in detail about how the severe disease affects the biomarkers in consideration, but this study tries to establish correlation of various factors with NAb titres. Disease severity is not satisfactorily liked with other parameters in the results section of the article.

A: Thank you again for your thorough review, insightful evaluation, and informative and valuable suggestions on this study. These comments have helped us to revise and improve the quality and context of our manuscript. After intensive discussion by the entire research team, we have addressed all comments and suggestions in our responses and have revised the manuscript and supplementary materials. We sincerely hope that these enhancements would address your questions and present the results clearer and more complete.

Reviewer 3 Report

The article needs to be checked and edited for some language errors.  Overall an easy read and gives a good summary of retrospective for data for Wuhan Covid-19 patients.  

Author Response

Comments and Suggestions for Authors

Q1: The article needs to be checked and edited for some language errors.  Overall an easy read and gives a good summary of retrospective for data for Wuhan Covid-19 patients. 

A: Thank you very much for your detailed review and careful checks of our manuscript. The manuscript has been polished by an English language editing company. It has been carefully reviewed by a native English researcher to continuously improve the quality of the English language and the readability of the manuscript. We have also carefully checked the whole manuscript to correct grammar/typo errors. Please find corrections in the revised manuscript.

Round 2

Reviewer 1 Report

Thanks for the revised version. The authors have improved the manuscript. However, they used (( )) too much. Can you please use [()] instead? 

Reviewer 2 Report

The authors have made significant improvements in both the writeup and presentation of manuscript. Also explanations were given to the queries raised.  No further comments on the revised draft.